# Metallophenolomics: A Novel Integrated Approach to Study Complexation of Plant Phenolics with Metal/Metalloid Ions

**DOI:** 10.3390/ijms231911370

**Published:** 2022-09-26

**Authors:** Volodymyr S. Fedenko, Marco Landi, Sergiy A. Shemet

**Affiliations:** 1Research Institute of Biology, Oles Honchar Dnipro National University, 72 Gagarin Avenue, 49010 Dnipro, Ukraine; 2Department of Agriculture, Food and Environment, University of Pisa, Via del Borghetto, 80I-56124 Pisa, Italy; 3Ukrainian Association for Haemophilia and Haemostasis “Factor D”, Topola-3, 20/2/81, 49041 Dnipro, Ukraine

**Keywords:** anthocyanins, binding sites, complexation, flavonoids, metal/metalloid, phenolic ligand

## Abstract

Plant adaptive strategies have been shaped during evolutionary development in the constant interaction with a plethora of environmental factors, including the presence of metals/metalloids in the environment. Among adaptive reactions against either the excess of trace elements or toxic doses of non-essential elements, their complexation with molecular endogenous ligands, including phenolics, has received increasing attention. Currently, the complexation of phenolics with metal(loid)s is a topic of intensive studies in different scientific fields. In spite of the numerous studies on their chelating capacity, the systemic analysis of phenolics as plant ligands has not been performed yet. Such a systematizing can be performed based on the modern approach of metallomics as an integral biometal science, which in turn has been differentiated into subgroups according to the nature of the bioligands. In this regard, the present review summarizes phenolics–metal(loid)s’ interactions using the metallomic approach. Experimental results on the chelating activity of representative compounds from different phenolic subgroups in vitro and in vivo are systematized. General properties of phenolic ligands and specific properties of anthocyanins are revealed. The novel concept of metallophenolomics is proposed, as a ligand-oriented subgroup of metallomics, which is an integrated approach to study phenolics–metal(loid)s’ complexations. The research subjects of metallophenolomics are outlined according to the methodology of metallomic studies, including mission-oriented biometal sciences (environmental sciences, food sciences and nutrition, medicine, cosmetology, coloration technologies, chemical sciences, material sciences, solar cell sciences). Metallophenolomics opens new prospects to unite multidisciplinary investigations of phenolic–metal(loid) interactions.

## 1. Introduction

The life processes of plants have evolved in coordination with environmental factors. In addition, intensified anthropogenic load on ecosystems has led to increasing levels of chemical contamination and resulted in the emergence of new pollutants, namely xenobiotics. To understand the peculiarities of plant–environment interactions, it is essential to take into account the environmental and ecological aspects of the problem [1]. Hazardous metals and metalloids are among the major and widespread pollutants due to their high toxicity to the biosphere and the amplitude of their contamination in the natural environment [2]. Mining and metal extraction, fossil fuel combustion, agricultural application of fertilizers, sewage sludge, metal-containing pesticides, wastewater irrigation, and atmospheric deposition are the main anthropic sources of metal(loid)s [3].

A special feature of plants regarding their relations to metal(loid)s is that a certain amount of trace metals is necessary for a number of biologically essential processes (metalloenzymes, mineral nutrition, photosynthesis, prooxidant/antioxidant systems, etc.), whilst both the overdoses of essential and toxic doses of non-essential elements negatively affect plant metabolism. Therefore, to optimize those processes, plants have evolved multiple regulatory and defence mechanisms to counteract metal(loid) toxicity [4,5,6].

Among others, plants detoxify metal(loid)s via their biotransformation into metabolically inactive compounds [7,8]. Among biotransformation reactions, the in vivo chelating of metal ions is pivotal, which is accomplished by endogenous chelators: glutathione (GSH), phytochelatins (PCs), metallothioneins (MTs), organic acids, nicotinamine, amino acids [6,9,10]. An important feature of those bioligands is their capacity to bind various metals [10]. Such a property of binding both essential and non-essential metal(loid)s has been confirmed in vivo for flavonoid pigments of anthocyanins (ACNs) [11,12,13]. Those results allow postulating a hypothesis about the participation of phenolic compounds (PCs) in metal(loid) detoxification in plants [12]. However, a comprehensive systemic analysis of the role of PCs as endogenous chelators in plant metal tolerance has not yet been performed.

It is noteworthy that the current level of metal(loid) tolerance in plants is characterized by the extensive use of the integrated “omics” approach [14,15,16,17,18]. The “omics” approach is aimed at the studying of the organism as a holistic system, based on the integrative analysis of and interrelations among major biological processes [19]. The “omics” research object related to the behaviour of metals in living organism is referred to as the “metallome” [20] and the corresponding research field as “metallomics” [21]. (These terms will be evaluated in detail in Section 2.1.)

In recent years, the key significance of PCs was confirmed as pivotal and versatile plant defensive compounds against abiotic stresses, including metal(loid) tolerance [5,22,23,24]. Systematizing extensive experimental data resulted in the hypothesis of a universal dominant tendency to increased accumulation of PCs as components of the antioxidant defence system, which ensures the balance between the production and detoxification of reactive oxygen species (ROSs) under metal(loid) exposure [25]. Beside their ROS scavenging prerogative, other possible roles of those secondary metabolites in the metallomics context remain a poorly investigated issue.

This review outlines (i) the basic concepts of metallomic studies, (ii) their differentiation into subgroups, (iii) the chemistry of the complexation of phenolic ligands with metal(loid)s, and (iv) the introduction to the novel integrated approach of metallophenolomics to study metal(loid)–phenolic interrelations.

## 2. Metallomics as a Scientific Approach

The need for the systemic evaluation of metal behaviours in living organisms has led to the development of new approaches for studying this problem. For plant–metal(loid) interactions, the most relevant is the concept of metallomics, which is reviewed in detail in the following subsections.

### 2.1. Basic Concepts of Metallomics

The new scientific field, namely “metallomics”, as an integrated biometal science was introduced by Haraguchi [24]; the historical aspect of the origin and development of metal-related omics approaches was systematized by the author in the review [26]. The definitions of the key terms and basic concepts of this scientific approach were described in [27].

The term “metallome” can be described as the entirety of metal and metalloid species present in a biological system, defined as their identity and/or quantity [27]. Firstly, the metallome is the distribution of metal ions in a cellular compartment. Secondly, the metallome is related to the definition of the total element concentrations, metallocomplexes with different ligands, or all the species of a certain element (in their free forms or included in eobiotic or xenobiotic molecules). To define the metal-bound compounds in living organisms, in addition to the term “bioligands”, the terms “metallobiomolecules” [28], “chelating agents” [29], and “chelators” [9] are used.

The entirety of the scientific branches studying the metallome is included in the broader definition of “metallomics”. The main characteristics of those studies are: (i) the focus on metals or metalloids in a biological context; (ii) the statistical, functional, or structural link between the set of the element concentration or element speciation and the genome; (iii) a systematic or comprehensive approach [27]. It should be noted that the main feature of metallomics is not only the identification of metals, but also the determination of their roles and effects in biological processes. A peculiarity of metallomics is the studies of the distribution of metal ions, their quantities, the multivariance of their interaction with different bioligands, and the spatial and temporal characteristics of this process during the development of the living organism, as well as the impact of genetic variability. The complicated nature of this science was characterized by [29] as “metametallomics”.

Metallomics is the modern scientific branch, which has demonstrated an active development in the recent decade; therefore, this concept has received much attention and was extensively evaluated by many authors in a number of reviews and monographs [29,30,31,32,33,34,35]. Metallomics is an interdisciplinary topic and merges different spheres of research to build a global and systemic understanding of metal-assisted functions in biological systems; it includes various scientific fields and research subjects [26]. For the systemic analysis of the functions of biological systems, researchers consider integrating metallomics with different other omics (metabolomics, genomics, transcriptomics) [36].

To characterize completely the metal-related omics approaches, it should be noted that, in addition to metallomics, other scientific directions have been proposed, which have related terminological definitions. Staring from 2003, when the concept of the *ionome* was firstly introduced by Lahner et al. [37], *ionomics* has been developed in parallel with metallomics, the science involving “quantitative and simultaneous measurements of the elemental composition of living organisms, and changes in this composition in response to physiological stimuli, developmental state, and genetic modifications” [38]. The further development of ionomics and its connection with other omics has recently been reviewed [18,39,40].

To expand the area of the omics approach and encompass all chemical elements present in living organisms, including non-metals, *elementomics* was introduced, as the “study of elements of interest and element species, and their interactions, transformations, and functions in biological systems” [41]. Of note, the omics study of a specific global effect of a particular metal is also defined; for example, As-induced stress in plants is defined as *arsenomics*, an integrated approach associated with transcriptomics, proteomics, and metabolomics [42,43,44]. The term “*metallometabolomics*” was introduced to characterize the whole entirety of metallo-metabolites or to identify some of the dominant metal-complexing metabolites, e.g., in metal-hyperaccumulating plants [31]. Zhang et al. [45] used the term “*elemental metabolomics*” for the “quantification and characterization of total concentration of chemical elements in biological samples and monitoring of their changes.” Another aspect in metal-related studies is defined as “*metallome homeostasis*”—the understanding of “how the individual metal homeostasis systems overlap and interact so that all required essential metals are obtained and routed to the correct locations” [46].

### 2.2. Differentiation of Metallomic Studies

The extensive development of the metallomic approaches in different scientific fields and research subjects has resulted in the accumulation of a vast volume of experimental data. Due to the need to systematize the available literature accumulating on the matter in different directions, a problem arises from the division of the separated subgroups in metallomic studies [47]. Metallomics could be divided into subgroups in several ways, depending on the criteria for classification, with the focus on various aspects of the problem.

*Specific experimental criteria*: Szpunar [30] suggested to distinguish *qualitative metallomics*, which is based on identifying individual metal species, and *quantitative metallomics*, which deals with metal concentrations. For the monitoring of metallome changes in time and under environmental factors, the term “*comparative metallomics*” can be used [30]. *Structural metallomics* studies the metal coordination environment in biological systems [29]. *Functional metallomics* investigates the role of metal ions in the functions of proteins (catalytic, structural, regulatory) and, in particular, enzymes (oxidoreductases, transferases, hydrolases, liases, isomerases, ligases) [29].

*Scientific field*: For plant sciences, the following directions/subgroups have been proposed. In general, for plant metallomics, the term *phytometallomics* is used [48]. The metallomic study of plants is viewed as a specific aspect of the directions with a broader scope, which have been proposed as independent interdisciplines. Thus, to study the problems related to metal(loid)s in agricultural science, the *agrometallomics* concept was created [49]. The environmental branch of metallomic study is termed *environmental metallomics* [50,51] or *envirometallomics* [52].

*Form of metal(loid) species*: Some subgroups within metallomics can be distinguished based on the different natures of various metal species. Thus, studies devoted to the behaviour of different metal isotopes are defined as *isotope metallomics* [53,54]. The metabolism and behaviours of radioactive elements are related to *radiometallomics* [55]. The extensive development of nanotechnology in recent years has resulted in the emergence of a new branch of metallomics—*nanometallomics*—which aims at quantitation, distribution, structural changes, metabolism, the elucidation of reactions and mechanisms of metal-related nanomaterials in biological systems, and specific nano-scaled metal(loid)-assisted function science in different fields [56]. Wang et al. [57] extended this approach to incorporate all nanomaterials and metal-biomolecular homeostasis processes and proposed *comparative nanometallomics* as a new tool for nanosafety evaluations.

*Nature of chemical elements*: Depending on the given element, the specific metallome subgroups were outlined, such as for iron [58,59], nickel [60], copper [61,62], zinc [63], and manganese [64].

*Structure of bioligand*: The best-studied subgroup within metallomics is metalloproteomics, which studies bioligands of a proteic nature [30,50,65,66,67]. Depending on the type of metal(loid) associated with the proteins, specific subgroups can be distinguished within metalloproteomics, devoted to better understanding the biological role of specific element, such as *selenoproteomics* for selenium [65]. The subgroup of metallomics that globally studies the thiol peptides and their metal complexes is defined as *metallothiolomics* [68,69]. Codd [70] introduced the term *metalloglycomics* for studying competitive metal-carbohydrate binding. This approach is associated with the search for new metallodrugs in bioinorganic chemistry for the systematic study of the interactions between metal ions and coordination compounds with oligosaccharides [71,72]. Currently, however, taking into account the modern data on oligosaccharides’ participation as high-molecular weight ligands in metal homeostasis in plants [73], metalloglycomics should be considered as a subgroup of metallomics.

The subgroup of metallomic studies dealing with the interaction between metals and lipids is designated as *metallolipidomics* [74].

Thus, the current development of metallomics is characterized by a high differentiation of this approach. However, the differentiation by the bioligand structure does not encompass a specific branch for plant chelators of a phenolic nature yet.

### 2.3. Potential Role of Phenolic Chelators in Plant Metallomic Studies

To establish the potential role of phenolic compounds (PCs), two important concepts of the metallomic approach must be considered. Firstly, according to this approach, the studies of the metal(loid)s’ behaviour in plants include the following aspects: (i) mobilization of low-soluble metals from soil; (ii) translocation within the plant; (iii) sequestration of metal ions in the cytosol or in cellular compartments [31]. In this regard, the interest from the researchers in low-molecular-weight metal(loid)-containing metabolites is increasing, due to the following reasons: (i) the uptake and bioavailability of essential elements, in particular Fe and Zn, are crucial for optimal elemental content in fruit and vegetables for human consumption; (ii) the plant’s ability to accumulate metals is the basis for phytoremediation technologies and for the screening of hyperaccumulators, which are able to accumulate high levels of elements from the environment; (iii) plant tolerance mechanisms toward the stress effects of metal(oid)s include the induction of endogenous chelators [31]. However, phenolics as metallo-molecules have not been considered yet in the simplified model of biological systems within the context of omics science [26]. Secondly, the note by Lobinsky et al. [27] is of crucial importance: “the description of metallome can never be complete”, which results from the multivariance of the process of metal complex formation with already-established or potential bioligands. Such an explication opens the prospect for the further development of the metallomics concept, through the investigation of metal binding properties in the metabolites, which previously had not been considered in the context of the metallome, e.g., PCs. 

## 3. Plant Phenolics as Ligands for Metal(loid)s

In our opinion, the systematization of the available information on the chelating capacity of plant PCs should be performed in two consecutive stages. In the first stage, it is necessary to analyse the binding of PCs with metal(loid)s’ ions in vitro to establish the structure of the metallocomplexes formed and the key criteria of such a binding. In the second stage, using the identified criteria of binding, it is possible to systematize the experimental results about the PC’s chelation with metal(loid) ions (Me^n+^) in plants in vivo.

### 3.1. Complexing In Vitro

In the studies of PC–Me^n+^ chelation, two main directions can be distinguished:(1)Evaluation of the complexation of individual PCs with Me^n+^, based on the features of the ligands, which are modified due to chelation;(2)Assessment of metal chelating ability toward PCs and plant extracts based on the alterations in the absorption of metallochromic indicators.

#### 3.1.1. Individual Phenolic Compounds

Various aspects related to the synthesis, identification of the structure, biological activity, and application of PC–Me^n+^ complexes have been systematized in numerous reviews [75,76,77,78,79,80,81,82]. However, some aspects of this problem remain unclear due to the scarce attention given to the involvement of PCs in the modulation of the metallome

In this regard, we analysed the available data on the ability of natural phenolic metabolites to form metallocomplexes or identifying the binding of individual compounds to Me^n+^ in vitro in order to answer the following questions:(1)How are metal binding properties manifested for the natural compounds from different PC subclasses, which are formed in the process of plant phenolic metabolism?(2)Which structural fragments of PCs are crucial for the complexation?(3)Can PCs be considered as universal ligands for multiple Me^n+^?

We systematized available experimental data, and the main results are presented in Table 1 for individual representatives of various plant PC subgroups. The structural formulas of the ligands are shown in Figure 1 with their division into separate subgroups (phenolic acids 1–12, coumarins 14–16, chalcones 17, dihydrochalcones 18, flavanones 19–23, flavanonols 24, 25, flavonols 26–37, flavan-3-ols 38–43, flavones 44–51, isoflavones 52–54, anthocyanidins 55–57, xanthonoids 58, stilbenes 59, curcuminoids 60, lignans 61, flavonolignans 62, lignins, tannins 63–65). For some flavonoids, the data on complexation are combined with their derivatives. The number of Me^n+^ ions, for which the formation of metal complexes is confirmed, is presented regardless of the compounds with different stoichiometric ligand:Me^n+^ ratios, or for one ligand with different Me^n+^ ions (heterometallic complexes), or for one Me^n+^ with different ligands (mixed complexes). For some ligands, radiolabelled complexes are included. Chemical elements, for which the complexation with PC ligands has been established, are provided in Figure 2. It should be noted that we analysed only the data on metal complexes with natural PCs; currently, however, numerous studies are being performed on synthetic ligands of a phenolic nature, obtained by structural modification of the binding sites of natural PCs in order to create new effective biologically active substances [77]. 

Phenolic acids, as the first structural subgroup of the plant PC metabolic pathway, could be divided into hydroxybenzoic and hydroxycinnamic acids depending on the direction of their biosynthesis [186]. Among natural hydroxybenzoic acids, the complexes with Me^n+^ have been identified for protocatechuic acid 1, vanillic acid 2, gallic acid 3, and syringic acid 4. Protocatechuic acid 1, depending on the pH, coordinates with Al(III) and U(VI) ions via the carboxyl group or the ortho-dihydroxyl group [83,84]. For vanillic acid 2, the complexation with 17 Me^n+^ ions has been identified (Table 1). For gallic acid 3, the coordination with Me^n+^ can involve the carboxylate and neighbouring phenolic hydroxyl groups [91]. The largest number of complexes with Me^n+^ (22 ions) among hydroxybenzoic acids was identified for syringic acid 4.

Transformation of cinnamic acid 5 in the shikimate pathway results in the formation of different hydroxycinnamic acids (*p*-coumaric acid 6, caffeic acid 7, ferulic acid 8, isoferulic acid 9, sinapic acid 10, chlorogenic acid 11, rosmarinic acid 12, chicoric acid 13) [186]. Cinnamic acid 5 forms complexes with 19 Me^n+^ ions using its carboxyl group. In the binding of *p*-coumaric acid 6 with Me^n+^, its hydroxyl group could additionally be involved. In caffeic acid 7, its *o*-dihydroxyl group as an additional chelating site increases the ability of this molecule for complexation. The ability to form metallocomplexes has been confirmed for their methoxy derivatives (ferulic acid 8, isoferulic acid 9, sinapic acid 10). The esters of caffeic acid with quinic acid (chlorogenic acid 11), dihydroxyphenyl-lactic acid (rosmarinic acid 12), and tartaric acid (chicoric acid 13) retain the capacity for complexation with Me^n+^.

For the following subgroups, the complexation with Me^n+^ has been exemplified by their representative compounds: *coumarins*—coumarin 14, umbellipherone 15, daphnetin 16; chalcones—butein 17; *dihydrochalcones*—phloretin 18 (Table 1, Figure 1).

*Flavanones* form metallocomplexes in the form of both aglycons (naringenin 19, eriodictyol 21, hesperitin 22) and glycosides (naringin 20, hesperidin 23). For flavanonols, the metal chelating capacity has been confirmed for their derivatives with catechol (taxifolin 24) and gallic (dihydromyricetin 25) fragments.

Among the most-studied PC bioligands are *flavonols* (kaempferol 26, quercetin 27, rutin 28, quercitrin 29, isoquercitrin 30, isorhamnetin 31, tamarixetin 32, fisetin 33, morin 34, myricetin 35, myricitrin 36, galangin 37) (Table 1, Figure 1). It is noteworthy that the greatest amount of coordinated metals (43 different Me^n+^ ions) has been identified for quercetin 27 and its glycosides (rutin 28, quercitrin 29, isoquercitrin 30). This pronounced capacity of quercetin to chelate metals is associated with its structural features, which determine the possibility of different variants for the interaction with Me^n+^. Thus, the quercetin molecule contains three potential binding sites (Figure 3): (1) between the 3-hydroxy and 4-carbonyl groups in the C ring; (2) between the 5-hydroxy (in A ring) and 4-carbonyl groups (in the C ring); (3) between the 3’- and 4’-hydroxy groups in the B ring [78].

Complexation of *flavan-3-ols* with Me^n+^ is carried out by catechol and gallic binding sites ((+)-catechin 38, its stereoisomer (-)-epicatechin 39, (+)-epigallocatechin 40, esters with gallic acid–(-)-epicatechin 3-gallate 41, (-)-epigallocatechin 3-gallate 42). In theaflavin 43, Me^n+^ binding may also involve its tropolone moiety [155].

For *flavones* (primuletin 44, chrysin 45, apigenin 46, luteolin 47, tricetin 48, baicalein 49, baicalin 50, acacetin 51) without 3-hydroxy groups in the C ring, the complexation with Me^n+^ may involve the binding sites between 5-hydroxy (in A ring) and 4-carbonyl (in the C ring) or the catechol and gallic moieties. In this subgroup, the greatest number of metallocomplexes was identified for chrysin 45 and luteolin 47 (each binds 10 various Me^n+^ ions).

*Isoflavone* ligands are represented by daidzein 52, genistein 53, and its O-methylated derivative biochanin A 54.

Metal chelating capacity has been demonstrated for *anthocyanins* and their glycosides with two or three hydroxyl groups in the B ring: cyanidin 55, delphinidin 56, petunidin 57. In contrast to other flavonoids, a specific peculiarity of ACNs is a pH-dependent dynamic equilibrium of aqueous solutions between several structural forms, which are capable of Me^n+^ binding [13]. Among these ligands, the greatest number of metal complexes was identified for cyanidin 55 and its glycosides (27 Me^n+^, in cationic and anionic forms).

For *xanthonoids*, metallocomplex formation was exemplified by mangiferin 58 (glucosylxanthone) and for *stilbenes* by resveratrol 59.

Among *curcuminoids*, the most comprehensively studied ligand is curcumin 60, which may bind 28 various Me^n+^ due to its capacity of keto-enol tautomerism.

The ability of *lignans* for complexation has been confirmed for secoisolariciresinol diglucoside 61 and of *flavonolignans* for silibinin 62 (10 Me^n+^ ions each).

The metal binding capacity of *lignin* as a polymeric phenol was studied for a ligno-cellulosic substrate with Mn(II), Cu(II), and Fe(III) ions (Merdy et al., 2003).

The presence of a great number of hydroxy groups in the structure of *tannins* (oligomeric and polymeric phenols) determines their high capacity for complexation with Me^n+^. This fact has been established for their different forms: condensed tannins (proanthocyanidins), oenothein B 63 (dimeric macrocyclic ellagitannin), ellagic acid 64, tannic acid 65. The latter is one of the most-studied PC ligands (21 Me^n+^).

Thus, our attempt at systematizing the available experimental results revealed that metallocomplexes can be formed by numerous representative ligands from 18 subgroups of plant PCs, and they are capable of binding 69 different Me^n+^ ions (63 chemical elements) in total (Figure 2).

#### 3.1.2. Metal Chelating Ability

The metal chelating ability is recognized as a generally accepted integrated indicator of the complexing capacity of PCs; it is used as one of the indicators in antioxidant assays [187]. The main aspects of this approach were summarized in the reviews [130,188]. The approach is based on the ability of selected metallochromic indicators to form complexes with Me^n+^, which absorb light in the visible wavelengths range. Upon the addition of the tested PC ligand, competitive binding with Me^n+^ occurs, with a subsequent decrease in the absorption, which is expressed as equivalents of standard chelators or the percentage metal chelating [130]. The binding ability of ligands could also be evaluated by stability constants [188]. For example, in Fe chelation, ferrozine and 2,2′-bipyridine are used as metallochromic indicators and EDTA and deferoxamine as standard metal chelators [130,189]. This approach enables the evaluation of the dependence between the structure of the PC ligand and its metal chelating activity; thus, a comparative analysis of this indicator extracts of medicinal plants is possible [189,190].

### 3.2. Chelating Effects In Vivo

In the studies of in vivo binding between phenolic metabolites and Me^n+^, two aspects should be highlighted: (1) production of blue anthocyanins (ACNs) in blue flowers and (2) the defensive role of chelation in plant tolerance to toxic metal exposure. It is noteworthy that the vast majority of the in vivo studies on this topic are devoted to ACNs as metal chelators. This is due to the fact of the availability of non-destructive methods for the binding identification based on the spectral characteristics of ACN-Me^n+^ complexes in plant tissues [13]. Blue flower coloration is associated with copigmentation of ACNs and the formation of pigment–copigment–Me^n+^ complexes (Yoshida et al., 2009). Copigmentation can be performed with and without the participation of metal ions [191]. Such studies could be systematized according to two directions, which differ in their levels of elucidation of the content and structural organization of the pigment complex. The first direction is the evaluation of various aspects of the formation of non-stoichiometric ACNs’ metallocomplexes, which are stabilized due to copigmentation with caffeoyl or coumaroyl derivatives of quinic acids or glycosylated flavonoids [192]. For *Hydrangea macrophylla*, *Phacelia campanularia*, and *Tulipa gesneriana* flowers, the major pigments of those complexes are the delphinidin glucosides, while the pigments of *Meconopsis grandis* flowers are primarily composed of cyanidin glucosides [192]. The binding with metal ions (Fe^3+^, Al^3+^, Mg^2+^) is considered as a necessary condition for the formation of those pigment complexes [192].

Another direction is systemic studies, which have resulted in the establishment of the unique structure of metalloanthocyanins. According to the term’s definition, metalloanthocyanin is a self-assembled, supramolecular complex metal-containing pigment, which comprises 6 ACN molecules, 6 flavone molecules, and 2 metal ions [192]. During blue flowers’ colour formation, three major mechanisms can be implemented, i.e., self-association, copigmentation, and metal complexation [192]. To date, the following metallochelates have been isolated and identified from blue flowers: protocyanin (*Centaurea cyanus*), commelinin (*Commelina communis*), protodelphin (*Salvia paterns*), cyanosalvianin (*Salvia uliginosa*), nemophilin (*Nemophila menziesii*) [192]. The constitutive components of those pigments are the ACNs having a chelating centre with two (cyanydin) or three (delphinidin) hydroxyls, flavonoid apigenin derivatives, and Mg^2+^ and Fe^2+^ ions [192]. In protocyanin, an additional coordination link of Ca^2+^ ions with flavone molecules has been established [192]. The advantages of the supramolecular structure for plants are the stability of the pigment complex at physiological pH and the increased tolerance to UV radiation, which play an important role in the implementation of the main function of ACNs during plant blooming under sun irradiation. The simultaneous presence of non-associated and chelated ACN molecules explains the phenomenon of purple coloration due to the mixing of the two colour stimuli, red and blue [13]. Different ratios between those ACN forms, when present in vivo, create various superpositions of their colour stimuli, thus resulting in colour variability with different hues of purple plant coloration, a feature that has an important evolutionary significance, as it allows a wide diversity of plant colours and better alignment with pollinators [193,194]. One peculiarity of metallo-anthocyanins is the ability to replace coordinated biogenic Me^n+^ with abiogenic Me ions, while the spectral characteristics of the metallocomplexes are retained. Thus, commelinin-like pigments can be formed by replacing Mg^2+^ with Cd^2+^, Zn^2+^, Co^2+^, Ni^2+^, and Mn^2+^ [192].

The ACNs’ capability of binding various Me^n+^ ions during the formation of pigment complexes in flowers allows hypothesizing that the chelating properties could be engaged for a different purpose—to decrease the toxicity of endogenous metals, thus increasing plant metal tolerance [195]. This hypothesis was confirmed using maize as a metal-excluder plant; the in vivo chelating effect of cyanidin-3-glucoside (Cya-3-glu) in maize root tissues was found for nine exogenous Me^n+^ (Mg^2+^, Fe^2+^, Cd^2+^, Ni^2+^, Pb^2+^, Al^3+^, VO^3−^, MoO_4_^2−^, Cr_2_O_7_^2−^) [11,196]. The reversible nature of Cya-3-glu–Pb^2+^ binding was found in maize roots, which can be controlled by manipulating the pH in the root solution [13]. An increase in the Pb^2+^ concentration in the root nutrient solution resulted in the increased formation of Cya-3-glu–Pb^2+^ complexes in maize roots in a dose-dependent manner [196].

The formation of ACN–metal complexes in the hypocotyls of *Brassica* plants was found upon their treatment with MoO_4_^2−^ and WO_4_^2−^ ion solutions [197,198].

The capability of binding Me^n+^ was also demonstrated for other PCs localized in various plant tissues. Thus, the study of ACNs’ distribution over the roots of *Lotus pedunculatus* Cav. confirmed the hypothesis about metal binding and detoxifying by proanthocyanins in plant vacuoles [199]. Al(III) metallocomplexes with epigallocatechin gallate and proanthocyanins were identified in the leaves, stems, and roots of *Camellia sinensis* [179] and an oenothein B (dimeric macrocycle ellagitannin) in the roots of *Eucalyptus camaldulensis* [180]. Cd^2+^ binding by polymerized polyphenols was demonstrated in the leaves of water plants [200]. According to Rocha et al. [201], the reduction of mercury toxicity in plants can be associated with the chelating activity of gallic acid.

The confirmation of the role of PC ligands in plant–metal homeostasis is the identification of the complexes of Cu(II) with quercetin, luteolin, and syringic acid in the berries of *Euterpe oleraceae* and *Vaccinum myrtyllus* [202].

Aluminium stimulates maize plants to secrete into the rhizosphere various endogenous PCs (catechin, catechol, quercetin) capable of complexing with Al^3+^, thus implementing one of the mechanisms of plant tolerance to the metal excess in the root nutrition medium [203]. The role of root-secreted coumarins was shown in iron-deficient plants by the acquisition of Fe through reduction and chelation [204,205]. It is noteworthy that the binding effects/capacity of the chelators with different structural groups (including PCs) by trace elements are considered as one of the mechanisms of the soil–plant interface [206]. In this relation, it should be highlighted that metal(loid)-induced accumulation of PCs by plants is associated with their protecting role in plant metal tolerance [25].

### 3.3. Properties of Phenolic Chelators

Summarizing the data on PC–Me^n+^ binding presented in Section 3.1 and Section 3.2 of this review, we can outline both general and specific properties of phenolic chelators. The general properties of different PC subgroups are the following:


1.*Presence of* Me^n+^
*binding sites with O atoms* of carbonyl, hydroxyl, or carboxylate groups: Depending on the phenolic subgroup and a number of OH- groups as substituents, different chelation variants are possible. Thus, for quercetin, Me^n+^ chelation can take place at three binding sites (Figure 3). For ACNs, Me^n+^ binding occurs due to two or three hydroxyl substitutions in the B ring [13]. The formation of chelate structures with an unsaturated cycle with two or three coordinated O atoms defines the stability of such metallocomplexes. However, the coordination of Me^n+^ with two O atoms of the carboxylate groups of PCs is also possible, e.g., in *p*-coumaric acid [207];2.*Universal affinity of PCs in relation to different* Me^n+^ in both cationic and anionic forms: During the systematizing of the experimental and literature data, we revealed that PC ligands from various subgroups form complexes with 69 Me^n+^ ions (63 chemical elements; Table 1, Figure 1). Binding to PCs is a characteristic of/universal for the elements differing in their roles in plants:Essential macronutrients (Ca, Mg, K) and essential micronutrients (Fe, Mn, Zn, Ni, Cu, B, Mo), which are necessary for the plant life cycle, cannot be substituted by other elements and are directly involved in plant metabolism [208];Beneficial elements (Al, Co, Na, Se) that promote the growth of various plant species, but are not essentially required for the completion of the plant life cycle [209,210];Non-essential metal(loid)s that are not involved in primary plant metabolism;Rare earth elements (REEs), which include the lanthanide group with 15 elements (Figure 2).The complexation of PC ligands with 11 elements essential for human life (Na, Mg, K, Ca, Mn, Fe, Co, Cu, Zn, Mo, Se) has also been confirmed [211]. Such universalism of PC ligands is due to the multi-elemental composition of the plant metallome. Thus, according to Watanabe et al. [212], in the leaves of species from different families of terrestrial plants, 42 chemical elements have been found. An additional argument for a close PC–Me^n+^ interrelation is the correlation between the plant accumulation of essential, beneficial, and non-essential elements with the content of total phenolics and total flavonoids [213];3.Formation of complexes with different Me^n+^:ligand molar ratios, depending on the nature of the metal ion, the structure of the phenolic chelator(s), and the pH [77,78];4.The phenolic ligands’ capacity to form complexes with multiple different Me^n+^. The examples of such chelates are the heterobimetallic complexes of quercetin–Cu–Sn_2_ and quercetin–Zn–Sn_2_ [214];5.The capability of certain PCs to form mixed ligand complexes and multiligand metal–phenolic assemblies. Thus, mixed ligand complexes have been obtained for Pt(II)–naringin–caffeic acid, Pt(II)–naringin–sinaptic acid, and complexes of V(V) with those ligands [106]. Porkodi and Raman [215] synthesized mixed complexes of curcumin and quercetin derivatives with Co(II), Ni(II), Cu(II), and Zn(II). This capability was employed for the fabrication of hybrid functional materials using metal–phenolic networks based on the polyphenol components of green tea infusions [216];6.Complexation of PC ligands with different metal(loid)s’ species (ions, oxides, isotopes, nanoparticles): In addition to Me^n+^, solid-phase chelation of flavonoids has been confirmed for Al_2_O_3_, SiO_2_, and TiO_2_ [217,218,219]. Radiolabelled complexes have been synthesized for ^99m^Tc with curcumin, rutin, and luteolin [139,220,221], and ^68^Ga with curcumin [139]. The chelating capacities of PCs are used in the “green chemistry” synthesis of biocompatible nanomaterials [222];7.The formation of metabolites with chelating capacity at all stages of the phenolic biosynthetic pathway: Such a property is manifested regardless of the features of a particular biosynthetic pathway, which are determined by the complexity of the interrelated and consequent transformations of metabolites from various subgroups and the specificity of the dominant metabolite accumulated in a particular plant species. According to the systematization of Me^n+^ complexation reactions undertaken, we identified 18 subgroups of PCs capable of metal(loid) binding (Table 1). This enables multivariant scenarios of Me^n+^ binding in plants, thus confirming the universal nature of phenolic chelators;8.Metal ions’ binding has been demonstrated for PCs localized in plant tissues with various functions and for endogenous metabolites secreted by plants into the rhizosphere. Thus, binding with Me^n+^ has been confirmed for PCs in roots [11,180,196,199], hypocotyls [197,198], stems [179], leaves [200], flowers [192], berries [202], and the rhizosphere [203,204,205];9.The chelating capacity of PCs is manifested in different physiological processes of plants. Phenolic metallocomplexes are engaged in photoreception and photoprotection [12], plant–pollinator interactions [12], antioxidant and prooxidant mechanisms [188], metal detoxifying in plant tissues and the rhizosphere [11,196,197,198,199,200,203], vacuolar sequestration [12], the mobilization and phytoavailability of deficient elements [204,205];10.The ability of plant PCs to modulate the metallome in animals: Thus, the use of hesperidin, naringin, and quercetin as dietary supplements results in an altered element profile [223,224].


Besides the aforementioned general properties inherent to the chelators from different phenolic subclasses, the specific ACN properties should be distinguished, which are due to the chemical structure of those compounds:*The phenomenon of pH-dependent dynamic equilibrium between different structural forms of ACNs*: Both unbound and chelated ACN forms could be present simultaneously in plant tissue, thus creating its colour variation: red–purple–blue [13];*Reversible nature of ACN binding with metal ions in plant tissue; the unbound ACN form can be regenerated by varying the pH*: Such a peculiarity has been established for the binding of Pb^2+^ with Cy-3-glu in maize roots [13]. This feature is based on the pH-dependent transformation of the ACN structure;*ACNs and copigmentation*: According to Trouillas et al. [191], copigmentation is defined as the formation (in the presence or absence of metal ions) of non-covalent complexes involving anthocyanin or anthocyanin-derived pigment(s) and a copigment(s), with the resulting changes in the optical properties of the pigment complex. Major natural copigments are the phenolic metabolites: hydrolysable tannins, flavonols, flavones, dihydroflavonols, flavanones, phenolic acids, and derivatives thereof [191]. ACNs themselves can act as copigments due to the self-association of the two molecules [191].

Specific features of ACNs define their unique properties as plant chelators due to the following points:(i)The ACN chromophore within the metallocomplex absorbs light in the UV and visible region, which provides the UV protection of plant tissue and attracts pollinators to flower petals. Binding with Me^n+^ increases can resistance to solar radiation;(ii)ACN copigmentation occurs with the participation of both PCs and chelators from other groups (organic acids, amino acids). Moreover, some ACNs and phenolic copigments contain malonyl, succinyl, and quinic fragments in their molecules [192]. Such structural features of ACNs and copigments increase the number of binding sites and modify the chelation activity;(iii)The ACNs’ capacity to biotransform biogenic (nutrient) elements and engage them in the processes of photoprotection and plant–pollinator interactions, which was formed during natural evolution, can be used by plants for the other functional role—detoxication of abiogenic metals as pollutants resulting from man-induced activities.

Therefore, the performed analysis of in vitro and in vivo complexing, along with the revealed general capacity of chelation of PCs from different subclasses and the specific properties of ACNs allow postulating that phenolic compounds comprise a separate group of plant chelators. The preceding systematization of the experimental results enables assessing the role of phenolic ligands in the context of metallomics.

## 4. Concept of Metallophenolomics

### 4.1. Definitions

According to the main concepts of metallomics, the analysis of the metallome, in addition to the study of metal(loid) content and their distribution between the cell compartments, also includes the determination of the complex formation process, which involves metal ions and a bioligand(s) [65]. In this respect, one of the variants of the differentiation of metallomics into its subgroups is based on the nature of the bioligand [65]. Our systematization of the data on the chelating capacities of PCs (Section 3.3 enables, analogous to the metalloproteome, the separation of the metallophenolome; the object of the latter is the interactions between phenolic ligands and metal ions. Accordingly, the corresponding branch related to the study of the metallophenolome should be defined as metallophenolomics. Metallophenolomics is a ligand-oriented subgroup of metallomics; thus, it represents an integrated approach to study the complexation of plant phenolics with metals/metal(loid)s in different research subjects.

This term we firstly introduced in previous publications [225,226] to describe the interactions between metals and plant PCs. It should also be noted that in the context of metabolomics, the following terms had already been proposed previously: phenol-omics [227], phenol metabolomics [228], lignomics [229].

### 4.2. Research Subjects

According to the main concepts of metallomics [26], metallophenolomics is an interdisciplinary area of research and relates to both basic (chemistry, botany, biology, medicine, pharmacy, agriculture), as well as applied science (food science, nutritional science, toxicology, health science, environmental/green science). It is reasonable to analyse the research area of metallophenolomics in relation to the research subjects that were defined by Haraguchi [26] for metallomics. Research subjects for metallophenolomics are provided in Figure 4:


(1)Quantitative distribution and imaging analysis of elements in plant tissues. The use of the chelating effect of PCs is exemplified by the determination of Al localization in plant tissues by complexing to morin as the fluorochrome with subsequent detection by confocal laser microscopy [230,231]. This technique complements histochemical assays, where chelating dyes are used, and metal localization is examined by light microscopy [232] or visible reflectance and tristimulus colorimetry [233]. Tannic acid can be employed as a natural chelator for labile iron imaging in the prevention and treatment of iron-associated cancer or other iron-overload disorders [234];(2)Speciation of elements in plants: To establish the chemical forms of metals in which they are bound to plant chelators, the general analytical approaches of metallomics can be employed, along with a set of specific methods based on the structural peculiarities of these bioligands.General approaches include universal techniques for analysing the metal-containing biomolecules, i.e., hyphenated techniques (e.g., HPLC-ICPMS/ESI-MS) [235]. *Hydrangea* blue complexes composed of 3-O-glycosyldelphinidin, Al^3+^, and 5-O-acilquinic acid were investigated by electrospray-ionization mass spectrometry [236]. However, some supplementary non-destructive methods for phenol bioligands’ investigation are available due to their following specific features. Firstly, some of those metabolites (e.g., anthocyanins) are in most cases localized in surface plant tissues. Secondly, the chromophore system in some phenolic metal chelating molecules defines selective light absorption in the visible region, thus enabling the use of non-destructive methods (e.g., reflectance spectroscopy, tristimulus colorimetry) based on the interaction of the light beam with pigmented plant tissues. Thirdly, the differences in spectral characteristics between the unbound and chelated forms of bioligands are the markers for identifying the in vivo binding. Such specific analytical techniques could be exemplified by identifying ACN binding to Me^n+^ in flowers and roots [13];(3)Structural analysis of metal binding by phenolic compounds: This research field includes investigating the structural features of PCs from various subclasses along with their chelation sites [77], the structure–activity relations (antioxidant and metal chelating properties) [237], in silico prediction of the binding sites’ structure, and the potential biological activities of metallocomplexes [238];(4)Elucidation of the reaction mechanisms of the metallophenolome using model phenol–metal complexes: Phenolic–Me^n+^ complexation can alter the antioxidant activity of PCs due to the reduction of transition metal ions and the induction of Fenton reactions [237]. Therefore, the iron or copper chelating properties of PCs are considered to be a variant of the antioxidative effect [188,189,239]. The antioxidant activity of some phenolic–metal complexes is superior to that of the parent ligands [77]. The effect of metal binding on the antioxidant activity of the molecule was studied in ferrous flavonoid mixtures [148]. An important element of redox processes in biological systems is also the prooxidant activity of phenolics, which is stimulated by Me^n+^ [240,241]. Flavonoid–metal complexes may exhibit superoxide dismutase activity, and their radical-scavenging activity is superior to the unbound flavonoid [242]. Phenolic–metal complexes are used in elucidating the reaction mechanisms with biologically important molecules such as DNA [243,244], proteins [243], pectins [245], and lipids [77]. Model phenolic metallocomplexes are employed in biomimetic studies of the mechanisms of action of metalloenzymes, which use flavonoids as a substrate [77];(5)Identification of unknown metallophenolics: An important problem in this research field remains the identification of stoichiometric and non-stoichiometric anthocyanin metallocomplexes in blue flowers [192] and phenolic–Me^n+^ complexes in plant tissues and the rhizosphere under the toxic impact of metals [13,203]. An example of such a novel approach is the identification of metal binding PC ligands in wine [246];(6)Targeted analysis of metal chelators in phenolic metabolism: Metabolomic profiling enables identifying the effects of various metals on the qualitative composition and quantitative content of individual phenolic metabolites as potential chelators. Thus, Cu’s effect on *Cucumber sativus* significantly upregulates 4-hydroxycinnamic acid compared to other phenolic metabolites [247]. The treatment of *Helianthus annuus* with Cr enabled identifying ten isocoumarin derivatives as target metal chelating compounds [248]. Different changes in the metabolic profiles of flavonols and hydroxicinnamic acids were identified for two metallicolous populations of *Arabidopsis halleri*, which demonstrated different mechanisms of Cr tolerance [249]. The higher tolerance of a red- versus a green-leafed cultivar of sweet basil against boron toxicity was hypothesized to be partially related to the capability of anthocyanins to act as B chelators [250,251,252,253];(7)Medical diagnosis of health and diseases related to trace metals: The chelating capacities of PCs are employed in numerous medical diagnostic techniques. Tannic acid may be applied for chelation and imaging of labile iron in iron-associated cancer or other iron-overload disorders [234]. The complex of morin with ^68^Ga was proposed as a novel radiopharmaceutical for diagnostic purposes and kidney cancer cell labelling [254]. The morin metal complexes with DNA were confirmed as an effective tool for the discrimination of anticancer drugs’ binding mechanism to DNA [255];(8)Metallodrug design: PC–Me^n+^ complexes demonstrate a broad spectrum of biological activities (anti-/pro-oxidant, antimicrobial, antiviral, anti-inflammatory, anti-diabetic, anticonvulsant, anticancer) [77,78,79,80]. An option to use various ligands differing in their binding sites and the universal property of chelating multiple Me^n+^ corroborates the prospects of PCs in metallodrug design. Flavonoid–Me^n+^ complexes are viewed as a novel class of therapeutical agents [76]. The complexation of PCs with Me^n+^ enables obtaining metallocomplexes with improved biological activity compared to their parent ligands [77,79]. An essential advantage of those complexes is the use of non-toxic natural chelators, thus reducing the potential toxicity of metallodrugs in chemotherapy [256];(9)Chemical evolution of the living systems and organisms on Earth: The evolution and sophistication of mechanisms for chelating different elements are deemed as a factor of the evolutionary development of organisms [257]. Plant systems’ evolution involves metal homeostasis networks [4,258,259]. One of the important evolutionary aspects of elemental hyperaccumulation is based on the involvement of multiple kinds of chelators in plant tolerance mechanisms [260]. The important evolutionary role of flavonoids with antioxidant and chelating capacities was shown in the adaptation of metallicolous populations, wherein divergent strategies were revealed for Cd uptake, translocation, and detoxifying in different genetic units of *Arabidopsis halleri* [249]. The chelation mechanism, which was formed during evolutionary development toward essential elements, is utilized in plants to detoxify non-essential metal(loid)s as pollutants from anthropogenic sources [13]. The universality of this tolerance mechanism was demonstrated in relation to a new class of contaminants—metal-containing nanoparticles (NPs) [222]. ACN–Me^n+^ complexation is the important factor of the colour evolution of flowers, which enhances plant polychroism in plant–pollinator interactions [192];(10)Mission-oriented biometal sciences. *Environmental science*: **One of the adaptive mechanisms in plant tolerance strategies against metal(loid) toxicity is deemed to be the chelation process involving different kinds of chelators [261,262]. The chelating properties of phenolic metabolites are integrated in the omics approach to study plant responses to metal stress (transcriptomics, proteomics, metabolomics, ionomics) [263]. To improve plant metal tolerance, various techniques are utilized, including pre-treatment by phenolic-rich plant extracts with chelating capacity [264,265].** Chelation processes are crucial in phytoremediation technologies for metal-contaminated areas [266]. The phytoremediation potency of plants depends on various factors including the content of phenolic chelators in plant tissues [267]. Chelating capacity determines the efficacy of phenolics as natural removing agents (biosorbents) from contaminated soils and wastewaters [268,269,270,271]. The defensive role of phenolic metabolites in plant responses to metal-containing NPs as a new class of contaminants has already been demonstrated [272,273,274].


*Food science and nutrition*: The chelating capacity of anthocyanins is utilized for their stabilization by metal ions and creating blue food colorants with health-promoting effects [275,276]. To enhance the photostability of ACNs as colorants, they are incorporated into mesoporous silica granules containing various metal ions [277]. ACN–Fe^3+^ complexation with coating was efficient in preserving anthocyanin pigments in thermally processed fruit products [278]. PCs with chelating and/or antioxidative effects are deemed as important food ingredients capable of preventing the accumulation of non-essential metals in the human body [279]. Due to this, phenol-rich plant species are extensively studied as potential sources of phytochemicals with high antioxidant and chelating activities [280,281,282]. Experimental elaborations of this issue comprise the basement for dietary strategies for the treatment of metal toxicity [283]. The human-health-promoting effects of PCs are used in food antioxidant applications [284], global flavonoid intake [285], food metabolome [286], foodomics [287], and nutrigenomics [288]. Ferric anthocyanin chelators increase the pH sensitivity of indicator films for monitoring freshness and preparing intelligent packaging of food [289].

*Medicine*: A broad spectrum of pharmacological activities determines the significance of phenolic–metal complexes as a novel class of therapeutic agents [76,79]. A key feature of the production of those drugs is the use of non-toxic natural chelators [256]. This peculiarity asserts the prospects of phenolic ligands versus synthetic chelators for metal chelation therapy [290,291]. Phenolics as food components with antioxidant and chelating activities are employed in developing preventive and therapeutic strategies for metal intoxication [292,293,294]. The chelating activity of flavonoids is appreciated as an important factor in multitarget-directed ligand strategies in the management of Alzheimer’s disease [295]. The novel direction of nanomedicine is associated with biocompatible flavonoid-mediated metal-containing nanomaterials with a dual function as both nano-carriers and nano-drugs in numerous medicinal applications [222].

*Cosmetology*: Owing to the antioxidant and chelating properties, plant phenolics are utilized as bioactive ingredients in the cosmetic industry [296]. A multicomponent powder of polyphenol-rich extracts was efficient in counteracting skin damages induced by metal deposition from air pollution in the environment [297]. Plant anthocyanins with chelating effects are used as renewable hair dyes that are free of any toxic effects of synthetic dyes [298]. 

*Coloration technology*: Metal chelating flavonoids are important natural colorants in textile dying [77]. To ensure affinity between textile materials and pigments, metal salts (Sn, Al, Fe, Cu, Cr) are needed as mordants [299]. The flavonoid–metal complexation in the dying process improved pigment stability and altered the colour characteristics [77]. By varying the flavonoid composition, combining different plant species, metal ions, and conditions of dying, striking colour diversity can be achieved [300]. The use of natural pigments represents an alternative to synthetic dyes in the development of sustainable and eco-friendly processes [301].

*Chemical science*: Flavonoid–metal complexation is utilized in preparative, analytical, and synthetic approaches. Thus, an ACN purification method was proposed, which is based on the ligand exchange mechanism and uses cationic resins charged with Fe(III) [302]. In quantitative analysis, flavonoid–metal chelation could be used in two ways. On the one hand, flavonoids are the chromogenic reagents for the determination of various metals by spectrophotometric or fluorometric detection [77,135]. On the other hand, flavonoid content in plant samples can be determined using metal ions as analytical reagents [77,303,304]. The capacity of PCs to reduce metal ions and stabilize them into nanoparticles by, i.e., chelation mechanism, comprises the basement for the green synthesis of metal-containing nanomaterials [305,306]. 

*Geochemistry*: An important role of phenolics in the complexation and reduction of Fe in the dissolution and transport of terrestrial iron to aquatic ecosystems was confirmed, being a key link in the global coupled iron and carbon cycles [307].

*Corrosion protection*: The capacity of flavonoids with catechol fragments to complex with Fe(III) plays a significant role in the corrosion inhibition mechanism [308]. PCs as ingredients of plant extracts determine the efficacy of environmentally sustainable and “green” corrosion inhibitors for metals and alloys [309].

*Material science*: Phenol–metal chelation is important for the creation of novel hybrid multifunctional biomaterials for chemical, biomedical, and environmental applications. Thus, direct gelation between tannic acid and metal ions produces metallogels, films, and capsules [310]. A versatile platform for the development of functional hybrid materials is the synthesis of coordination-driven assemblies of metal–phenolic networks [311]. To create new biopolymers, a formaldehyde-assisted metal–ligand crosslinking strategy was proposed for the synthesis of metal–phenolic coordination spheres based on the principles of sol–gel chemistry [312]. A nanostructured porous carbon monolith was obtained based on phenolic–metal interactions (tannic acid–Zn chloride) as sorbent multi-scale molecules. The resulting material provides versatile adsorption behaviours ranging from small gas molecules to larger molecules such as dyes, oils, and organic solvents [313]. A natural antibiotic system was developed using tannic acid–metal coordination coating of curcumin nanoparticles [314].

*Solar cell science*: One of the most promising renewable and sustainable technologies is the dye-sensitized solar cell technology, an environmentally friendly method for producing electricity [315,316]. Among natural dyes from various plants, flavonoids and, in particular, ACNs are crucial, due to their ability to chelate TiO_2_ NPs [317,318]. 

Therefore, the research area of metallophenolomics includes major research subjects that have been defined for metallomics [21,26] and corroborates the multidisciplinary nature of this scientific direction [32,211]. 

## 5. Conclusions

The polyfunctionality of phenolic compounds (PCs) determines their important adaptive role in plant responses to metal(loid) stress and tolerance. Binding sites in the molecules of phenolics from various subclasses underlie their chelating capacity toward different metal(loid)s, metal(loid) oxides, and metal(loid)-containing nanoparticles. The chelating effect is exhibited by phenolic compounds localized in various plant tissues and organs (roots, leaves, hypocotyls, flowers), as well as the rhizosphere. Phenolic–metal(loid) complexation is pivotal for multiple plant processes: photoreception and photoprotection, plant–pollinator interactions, antioxidant and prooxidant mechanisms, metal detoxifying in plant tissues and the rhizosphere, vacuolar sequestration, and mobilization of elements and phytoavailability. Among the PCs, the structural features of anthocyanins determine their specific properties as phenolic chelators, which is associated with the dynamic equilibrium between their different structural forms and copigmentation effects. An integrated approach studying metal(loid)–phenolic complexation is described as metallophenolomics, a ligand-oriented subgroup of the field of metallomics. The concept of metallophenolomics opens a novel scientific route of research, which enables joining the efforts of the scientific community in multidisciplinary investigations of phenolic–metal interactions and metal-assisted areas of applications.

## Figures and Tables

**Figure 1 ijms-23-11370-f001:**
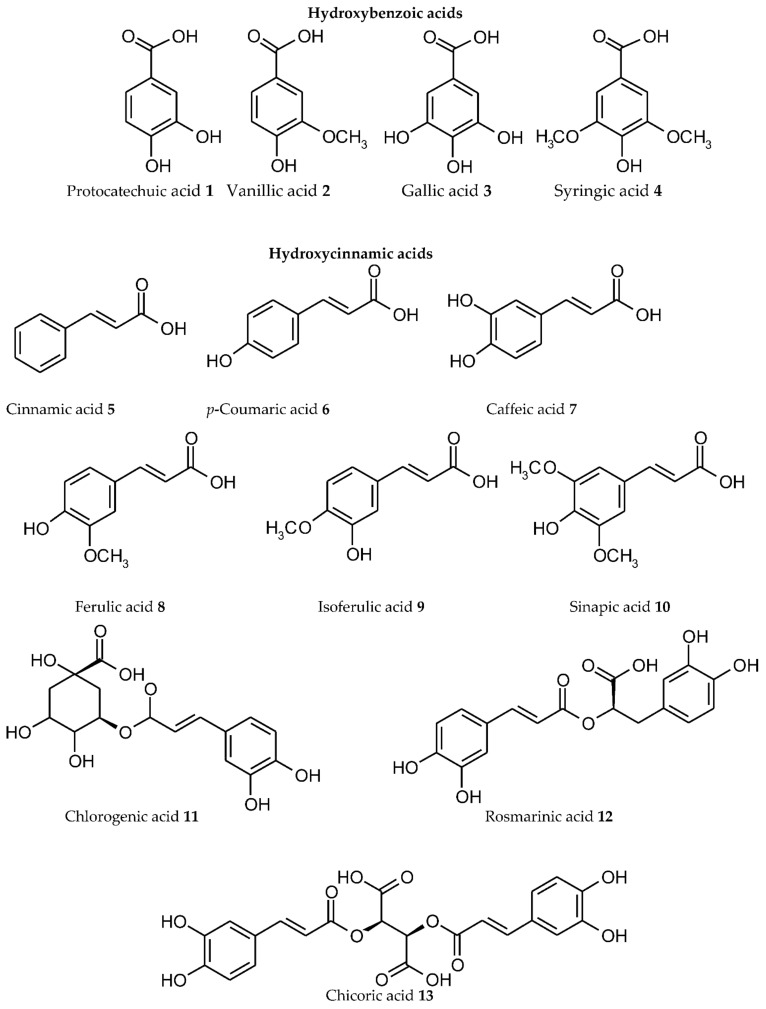
Structure of individual representative ligands capable of binding metal/metalloid ions from various plant phenolic subgroups.

**Figure 2 ijms-23-11370-f002:**
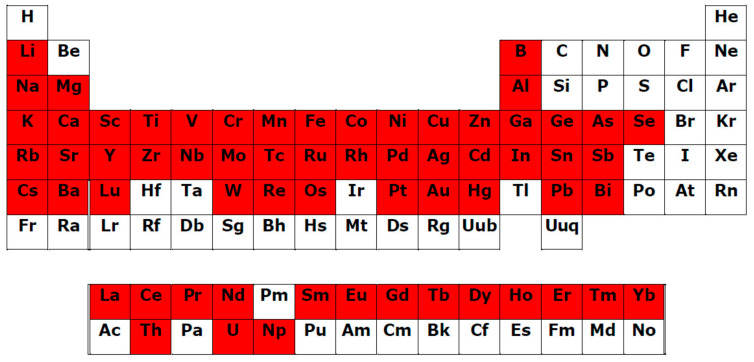
The elements confirmed to form phenolic ligand–Me^n+^ complexes (highlighted in red).

**Figure 3 ijms-23-11370-f003:**
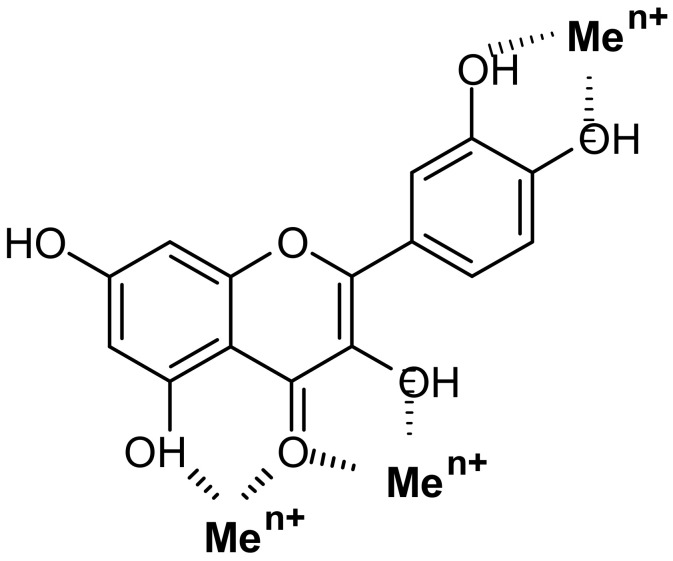
Possible binding sites of quercetin according to [77,78].

**Figure 4 ijms-23-11370-f004:**
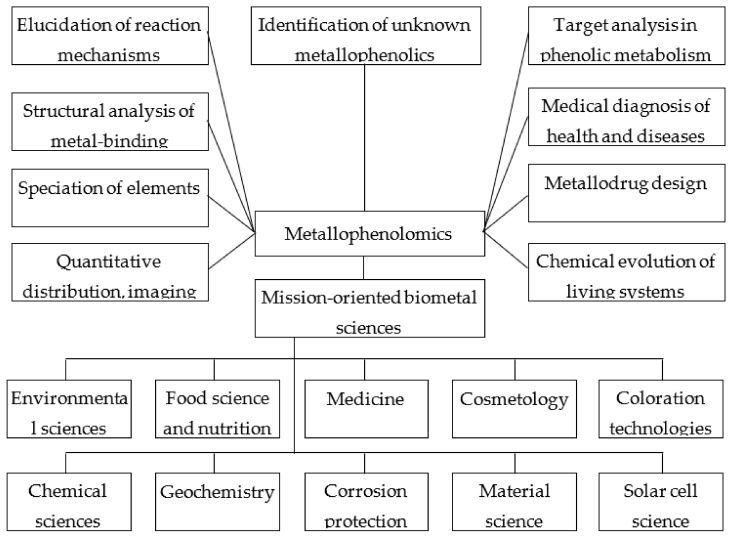
Research subjects of metallophenolomics.

**Table 1 ijms-23-11370-t001:** Complexes of plant phenolic ligands with metal(loid) ions.

Phenolic Ligand	Metal(loid) Ion	Number of Metal Ions	References
Phenolic acid			
*Hydroxybenzoic acids*			
Protocatechuic acid 1	Al(III), U(VI)	2	[83,84]
Vanillic acid 2	Zn(II), Y(III), La(III), Ce(III), Pr(III), Nd(III), Sm(III), Eu(III), Gd(III), Tb(III), Dy(III), Ho(III), Er(III), Tm(III), Yb(III), Lu(III), Np(V)	17	[85,86,87]
Gallic acid 3	Fe(II), Zn(II), Fe(III), Eu(III)	4	[88,89,90,91]
Syringic acid 4	Li(I), Na(I), K(I), Rb(I), Cs(I), Fe(II), Fe(III), Y(III), La(III), Ce(III), Pr(III), Nd(III), Sm(III), Eu(III), Gd(III), Tb(III), Dy(III), Ho(III), Er(III), Tm(III), Yb(III), Lu(III)	22	[92,93,94]
*Hydroxycinnamic acids*			
Cinnamic acid 5	Li(I), Na(I), K(I), Rb(I), Cs(I), Hg(I), Ca(II), Co(II), Ni(II), Cu(II), Zn(II), Ru(II), Cd(II), La(III), Eu(III), Tb(III), VO(IV), Th(IV)	19	[95,96,97,98,99,100,101,102]
*p*-Coumaric acid 6	Li(I), Na(I), K(I), Rb(I), Cs(I), Co(II), Ni(II), Cu(II), Zn(II),Al(III)	10	[82,103,104]
Caffeic acid 7	Li(I), Na(I), K(I), Rb(I), Cs(I), Cu(II), Pb(II), Pt(II), Al(III), Fe(III), Cr(III), Eu(III)	12	[82,105,106,107,108]
Ferulic acid 8	Ca(II), Mn(II), Cu(II), Zn(II), Cd(II), Al(III), VO(IV), V(V)	8	[61,82,97,109]
Isoferulic acid 9	Na(I), Mg(II), Mn(II)	3	[110]
Sinapic acid 10	Cu(II), Pt(II), V(V)	3	[106,111]
Chlorogenic acid 11	Li(I), Na(I), K(I), Rb(I), Cs(I), Ca(II), Zn (II), Fe(III), VO(IV)	9	[112,113,114,115,116]
Rosmarinic acid 12	Li(I), Na(I), K(I), Rb(I), Cs(I), Ca(II), Cu(II)	7	[115,117,118]
Chicoric acid 13	Co(II), Ni(II), Cu(II), Zn(II)	4	[119]
Coumarins			
Coumarin 14	La(III), Ce(III), Nd(III), Sm(III), Dy(III)	5	[108]
Umbellipherone 15	Ce(III)	1	[120]
Daphnetin 16	Cu(II), Zn(II), Ge(IV)	3	[121]
Chalcones			
Butein 17	Cu(II), Zn(II)	2	[122]
Dihydrochalcones			
Phloretin 18	Ru(III)	1	[123]
Flavanones			
Naringenin 19, naringin 20	Fe(II), Cu(II), Ni(II), Zn(II), Pt(II), Fe(III), Cr(III), La(III), Y(III), Eu(III), Ce(IV), VO(IV), V(V)	12	[76,77,106,124,125,126]
Eriodictyol 21	Fe(II), Fe(III)	2	[127]
Hesperitin 22, hesperidin 23	Ni(II), Cu(II), Zn(II), Al(III), VO(IV),	5	[76]
Flavanonols			
Taxifolin 24	Fe(II), Ni(II), Cu(II), Zn(II), Fe(III)	5	[77,124,128,129,130]
Dihydromyricetin 25	Mn(II), Fe(II), Co(II), Ni(II), Cu(II), Zn(II)	6	[131,132]
Flavonols			
Kaempferol 26	Fe(II), Cu(II), Zn(II), Pb(II), Fe(III), VO(IV)	6	[76,77,81,133]
Quercetin 27, rutin 28, quercitrin 29, isoquercitrin 30	Mg(II), Ca(II), Sc(II), Mn(II), Fe(II), Co(II), Ni(II), Cu(II), Zn(II), Mo(II), Pd(II), Cd(II), Hg(II), Sn(II), Pb(II), Al(III), Cr(III), Fe(III), Ga(III), Y(III), Rh(III), Sb(III), La(III), Pr(III), Nd(III), Eu(III), Gd(III), Tb(III), Dy(III), Tm(III), Au(III), Ge(IV), Zr(IV), Ru(IV), Sn(IV), Os(IV), Cr(VI), Mo(VI), W(VI), Tc(VII), Os(VIII), VO(IV), UO_2_(II),	43	[76,77,78,81,134,135,136,137,138,139,140]
Isorhamnetin 31	Fe(II), Cu(II)	2	[141]
Tamarixetin 32	Fe(II), Cu(II)	2	[141]
Fisetin 33	Fe(II), Cu(II), Zn(II), Fe(III), VO(IV)	4	[77,142]
Morin 34	Mg(II), Ca(II), Mn(II), Co(II), Ni(II), Cu(II), Zn(II), Sr(II), Pd(II), Cd(II), Ba(II), Sn(II), Pt(II), Al(III), Cr(III), Fe(III), Au(III), La(III), Eu(III), Gd(III), Lu(III), Zr(IV), VO(IV), Mo(VI), W(VI), Ti(COO)_2_ ^2+^	26	[76,77,78,143]
Myricetin 35, myricitrin 36	Cu(II), Zn(II), Al(III), Fe(III)	4	[76,78,136]
Galangin 37	Fe(II), Cu(II), Zn(II), Al(III)	4	[133]
Flavan-3-ols			
(+)-Catechin 38, (-)-epicatechin 39	Fe(II), Cu(II), Zn(II), Hg(II), Al(III), Fe(III), Cr(III), La(III), Yb(III), Gd(III)	10	[77,144,145,146,147,148,149,150,151]
(+)-Epigallocatechin 40	Fe(II)	1	[148]
(-)-Epicatechin 3-gallate 41	Fe(II), Cu(II), Zn(II), Al(III), Fe(III)	3	[80,146]
(-)-Epigallocatechin 3-gallate 42	Fe(II), Mn(II), Cu(II), Zn(II), Pt(II), Al(III), Fe(III)	7	[80,146,148,152,153]
Theaflavin 43	Al(III), Fe(III)	2	[154,155]
Flavones			
Primuletin 44	Zn(II), Cu(II); Pb(II), Al(III), Fe(III)	5	[77,133]
Chrysin 45	Cu(II), Pd(II), Al(III), Fe(III), La(III), Ho(III), Er(III), Yb(III), Ce(IV), VO(IV)	10	[76,77,133]
Apigenin 46	Cu(II), Pb(II), VO(IV)	3	[76,133,156]
Luteolin 47	Mn(II), Fe(II), Cu(II), Al(III), Fe(III), Y(III), Ho(III), Yb(III), Lu(III), VO(IV)	10	[76,77,81,156,157]
Tricetin 48	Fe(II), Fe(III)	2	[127]
Baicalein 49, baicalin 50	Fe(II), Cu(II), Fe(III), VO(IV)	4	[76,77,156]
Acacetin 51	Fe(III)	1	[158]
Isoflavones			
Daidzein 52	Ce(IV)	1	[77]
Genistein 53	Cu(II), Fe(III)	2	[76,159]
Biochanin A 54	Cu(II), Fe(III)	2	[159]
Anthocyanidins			
Cyanidin 55 and its glycosides	Cs(I), Mg(II), Ca(II), Mn(II), Fe(II), Co(II), Ni(II), Cu(II), Sr(II), Zn(II), Cd(II), Sn(II), Ba(II), Hg(II), Pb(II), B(III), Al(III), V(III), Cr(III), Fe(III), Ga(III), As(III), Bi(III), Ge(IV), VO_3_^−^, MoO_4_^2−^, WO_4_^2−^	27	[13,160,161,162,163]
Delphinidin 56 and its glycosides	Mg(II), Zn(II), Sn(II), Al(III), Cr(III), Fe(III), Ga(III)	7	[13,163,164]
Petunidin 57 and its glycosides	Mg(II), Sn(II), Al(III), Cr(III), Fe(III), Ga(III)	6	[164,165]
Xanthonoids			
Mangiferin 58	Fe(II), Cu(II), Zn(II), Fe(III), Se(IV), Ge(IV)	6	[121,166,167]
Stilbenes			
Resveratrol 59	Fe(II), Cu(II), Zn(II), Al(III), Fe(III)	5	[80,130,168,169]
Curcuminoids			
Curcumin 60	Mg(II), Ca(II), Mn(II), Fe(II), Co(II), Ni(II), Cu(II), Zn(II), Se(II), Pd(II), Cd(II), Sn(II), Hg(II), Pb(II), Al(III), Cr(III), Fe(III), Ga(III), Y(III), Ru(III), In(III), Re(III), Sm(III), Eu(III), Dy(III), Au(III), VO(IV), Nb(V)	28	[130,170,171,172]
Lignans			
Secoisolariciresinol diglucoside 61	Ag(I), Ca(II), Fe(II), Ni(II), Cu(II), Pb(II)	6	[173]
Flavonolignans			
Silibinin(silybin) 62	Ni(II), Cu(II), Zn(II), Fe(III), Ga(III), VO(IV)	6	[156,174,175,176]
Lignins			
Ligno-cellulosic substrate	Mn(II), Cu(II), Fe(III)	3	[177]
Tannins			
Condensed tannins	Fe(II), Cu(II), Zn(II), Al(III)	4	[178,179]
Oenothein B 63	Al(III)	1	[180]
Ellagic acid 64	Mg(II), Ca(II), Mn(II), Fe(II), Co(II), Cu(II), Fe(III)	7	[181]
Tannic acid 65	Mg(II), Mn(II), Fe(II), Co(II), Ni(II), Cu(II), Zn(II), Mo(II), Cd(II), Al(III), V(III), Cr(III), Fe(III), Ru(III), Rh(III), Ce(III), Eu(III), Gd(III), Tb(III), Ti(IV), Zr(IV)	21	[153,182,183,184,185]

## Data Availability

Not applicable.

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
