# Peer review of "Metallophenolomics: A Novel Integrated Approach to Study Complexation of Plant Phenolics with Metal/Metalloid Ions"

_ijms, 2022, doi:10.3390/ijms231911370_

Round 1

Reviewer 1 Report

The manuscript by Fedenko et al. is a sound critical review on metallophenolomics as a novel integrated approach to study complexation of plant phenolics with metal/metalloid ions. This is a frontier topic in plant science and deserve attention for both basic and applicative science.

 Overall the manuscript is well written, clear, exhaustive and easily understandable. In particular, the authors operated such a deep review survey (more than 300 references are reported and results depicted) which clearly supported a pivotal role of flavonoids and in particular anthocyanins as possible in planta metal and metalloid chelators.

The part related to the possible exploitation of phenolic-metal and metalloids is also very interesting for a broad audience, thereby making conceivable that the manuscript will be highly cited, even in a short future.

Below, I just want to describe some minor points that the authors may want to follow meanwhile revising their manuscript:

Fig. 1: The figure is quite important for the understanding of the manuscript topic, however has to be improved (e.g. standardizing the compound size, etc…)

Fig. 1 caption. It could be improved in order to better description of the Fig. material

Fig. 2 caption: it should be refreshed and improved

Line 346: What is Savia patens?

377-378: please refresh the first part of the sentence

Fig. 3: The quality has to be improved

396: p-coumaric (p should be italic; for here and elsewhere)

548: Arabidopsis halleri not hallery (as per line 570 correctly mentioned)

612-629: “coloration science” and “corrosion science” seems very strange terms. Please refresh this sentence

Author Response

Dear Editor,

besides going into detailed response to referee comments (in bold), we want express our sincere gratitude to the referees for their valuable comments, which have increased the scientific level as well as the readability of our manuscript. We hope that now the manuscript could be accepted for publication.

Referee 1

The manuscript by Fedenko et al. is a sound critical review on metallophenolomics as a novel integrated approach to study complexation of plant phenolics with metal/metalloid ions. This is a frontier topic in plant science and deserve attention for both basic and applicative science.

 Overall the manuscript is well written, clear, exhaustive and easily understandable. In particular, the authors operated such a deep review survey (more than 300 references are reported and results depicted) which clearly supported a pivotal role of flavonoids and in particular anthocyanins as possible in planta metal and metalloid chelators.

The part related to the possible exploitation of phenolic-metal and metalloids is also very interesting for a broad audience, thereby making conceivable that the manuscript will be highly cited, even in a short future.

Our reply: we are delighted from the referee comments

Below, I just want to describe some minor points that the authors may want to follow meanwhile revising their manuscript:

Fig. 1: The figure is quite important for the understanding of the manuscript topic, however has to be improved (e.g. standardizing the compound size, etc…)

Our reply: thank you very much. We completely redraw all the compounds formula. In the revised version, those are completely editable from the editorial staff to ensure the best quality and standardization once published.

Fig. 1 caption. It could be improved in order to better description of the Fig. material

Our reply: done.

Fig. 2 caption: it should be refreshed and improved

Our reply: done.

Line 346: What is Savia patens?

Our reply: thank you very much for pointing out the mistake in the name Salvia

377-378: please refresh the first part of the sentence

Our reply: done.

Fig. 3: The quality has to be improved

Our reply: the figure has completely been redrawn.

396: p-coumaric (p should be italic; for here and elsewhere)

Our reply: done.

548: Arabidopsis halleri not hallery (as per line 570 correctly mentioned)

Our reply: thank you very much for pointing out the mistake.

612-629: “coloration science” and “corrosion science” seems very strange terms. Please refresh this sentence

Our reply: thank you very much we changed those terms.

Reviewer 2 Report

The review paper focuses on the interactions of plant phenolics with metals/metalloids present in the environment, using the metallomic approach. The topic of regulatory and defense mechanisms in plants is important in terms of counteracting the toxicity.  

The manuscript has been well presented. All the review objectives have been achieved. The scope of issues raised is very wide.   

Because phenolic compounds embrace numerous subgroups and compounds (as shown in table 1), my question is if it is possible to conclude about the chelating activity of different phenolic subgroups not for individual compounds, as given in Table 1, but taking into consideration their groups based on their chemical structure.

In addition, is the fact that phenolic compounds are present in plants in the forms bound with other compounds, e.g. lignins, important for the subjected interactions? This was mentioned in lines 299-300; however, it would be interesting to expand it.

After answering the above-mentioned questions, I recommend accepting the manuscript.

Author Response

Dear Editor,

besides going into detailed response to referee comments (in bold), we want express our sincere gratitude to the referees for their valuable comments, which have increased the scientific level as well as the readability of our manuscript. We hope that now the manuscript could be accepted for publication.

Referee 2

The review paper focuses on the interactions of plant phenolics with metals/metalloids present in the environment, using the metallomic approach. The topic of regulatory and defense mechanisms in plants is important in terms of counteracting the toxicity. 

The manuscript has been well presented. All the review objectives have been achieved. The scope of issues raised is very wide.  

Our reply: we are thankful for such appreciations on our manuscripts.

Because phenolic compounds embrace numerous subgroups and compounds (as shown in table 1), my question is if it is possible to conclude about the chelating activity of different phenolic subgroups not for individual compounds, as given in Table 1, but taking into consideration their groups based on their chemical structure.

Our reply: In our opinion, the mentioning of individual representatives is necessary, since the process of complex formation is determined not only by the skeleton of subgroups, but also by the presence of various substituent, functional groups capable of creating further binding sites (for example, catechol fragment).

In addition, is the fact that phenolic compounds are present in plants in the forms bound with other compounds, e.g. lignins, important for the subjected interactions? This was mentioned in lines 299-300; however, it would be interesting to expand it.

Our reply: The comments raised by the referee is quite important. We also asked ourselves the importance of such polyphenols-based polymers for metal interaction. However, only few studies have investigated the topic to allow any generalization and we do not want to speculate too much. We prefer to wait future research on matter to understand better (and likely confer consistency to the already available results) the relationship between lignin/metal(loid) in planta. We are really delighted for such a comment because it reveals the high scientific level of the revision.

After answering the above-mentioned questions, I recommend accepting the manuscript.
